# P2Y12 Inhibitor Monotherapy versus Conventional Dual Antiplatelet Therapy in Patients with Acute Coronary Syndrome after Percutaneous Coronary Intervention: A Meta-Analysis

**DOI:** 10.3390/ph16020232

**Published:** 2023-02-03

**Authors:** Wen-Han Feng, Yong-Chieh Chang, Yi-Hsiung Lin, Hsiao-Ling Chen, Chun-Yin Chen, Tsung-Han Lin, Tzu-Chieh Lin, Ching-Tang Chang, Hsuan-Fu Kuo, Hsiu-Mei Chang, Chih-Sheng Chu

**Affiliations:** 1Department of Internal Medicine, Kaohsiung Municipal Ta-Tung Hospital, Kaohsiung Medical University Hospital, Kaohsiung Medical University, Kaohsiung 801, Taiwan; 2Institute of Clinical Medicine, National Cheng Kung University Hospital, College of Medicine, National Cheng Kung University, Tainan 704, Taiwan; 3Department of Pharmacy, Kaohsiung Municipal Ta-Tung Hospital, Kaohsiung 801, Taiwan; 4Division of Cardiology, Department of Internal Medicine, Kaohsiung Medical University Hospital, Kaohsiung Medical University, Kaohsiung 807, Taiwan; 5Center for Lipid Biosciences, Kaohsiung Medical University Hospital, Kaohsiung 807, Taiwan; 6Regenerative Medicine and Cell Therapy Research Center, Kaohsiung Medical University, Kaohsiung 807, Taiwan; 7Institute of Health and Welfare Policy, National Yang Ming Chiao Tung University, Taipei 112, Taiwan

**Keywords:** P2Y12 inhibitor monotherapy, percutaneous coronary intervention (PCI), acute coronary syndrome (ACS)

## Abstract

P2Y12 inhibitor monotherapy is a feasible alternative treatment for patients after percutaneous coronary intervention (PCI) in the modern era. Clinical trials have shown that it could lower the risk of bleeding complications without increased ischemic events as compared to standard dual antiplatelet therapy (DAPT). However, the efficacy and safety of this novel approach among patients with acute coronary syndrome (ACS) are controversial because they have a much higher risk for recurrent ischemic events. The purpose of this study is to evaluate the efficacy and safety of this novel approach among patients with ACS. We conducted a meta-analysis of randomized controlled trials that compared P2Y12 inhibitor monotherapy with 12-month DAPT in ACS patients who underwent PCI with stent implantation. PubMed, Embase, the Cochrane library database, ClinicalTrials.gov, and other three websites were searched for data from the earliest report to July 2022. The primary efficacy outcome was major adverse cardiovascular and cerebrovascular events (MACCE), a composite of all-cause mortality, myocardial infarction, stent thrombosis, or stroke. The primary safety outcome was major or minor bleeding events. The secondary endpoint was net adverse clinical events (NACE), defined as a composite of major bleeding and adverse cardiac and cerebrovascular events. Five randomized controlled trials with a total of 21,034 patients were included in our meta-analysis. The quantitative analysis showed a significant reduction in major or minor bleeding events in patients treated with P2Y12 inhibitor monotherapy as compared with standard DAPT(OR: 0.59, 95% CI: 0.46–0.75, *p* < 0.0001) without increasing the risk of MACCE (OR: 0.98, 95% CI: 0.86–1.13, *p* = 0.82). The NACE was favorable in the patients treated with P2Y12 inhibitor monotherapy (OR: 0.82, 95% CI: 0.73–0.93, *p* = 0.002). Of note, the overall clinical benefit of P2Y12 inhibitor monotherapy was quite different between ticagrelor and clopidogrel. The incidence of NACE was significantly lower in ticagrelor monotherapy as compared with DAPT (OR: 0.79, 95% CI: 0.68–0.91), but not in clopidogrel monotherapy (OR: 1.14, 95% CI: 0.79–1.63). Both clopidogrel and ticagrelor monotherapy showed a similar reduction in bleeding complications (OR: 0.46, 95% CI: 0.22–0.94; OR: 0.60, 95% CI: 0.44–0.83, respectively). Although statistically insignificant, the incidence of MACCE was numerically higher in clopidogrel monotherapy as compared with standard DAPT (OR: 1.50, 95% CI: 0.99–2.28, *p* = 0.06). Based on these findings, P2Y12 inhibitor monotherapy with ticagrelor would be a better choice of medical treatment for ACS patients after PCI with stent implantation in the current era.

## 1. Introduction

Based on the results of CURE study and concerning stent thrombosis with first-generation drug-eluting stents (DES), 12-months of dual antiplatelet therapy (DAPT) has been the standard care for ACS patients after percutaneous coronary intervention (PCI) and stent implantation for the last few decades [1,2]. Although DAPT could reduce the risk of ischemic events, it also increases the risk of bleeding complications. These bleeding events used to be assumed to be benign, but recent studies had shown that post-PCI bleeding was associated with a substantial risk of recurrent ischemic events and increased mortality [3,4]. With the advent of newer-generation DESs and the advancement of PCI techniques, the risk of stent thrombosis is much lower than before. Some researchers had challenged this standard treatment strategy by trying to shorten the duration of DAPT with the continuation of aspirin monotherapy. Although some studies had shown the safety of this approach, they generally enrolled predominantly low-risk patients and excluded ACS patients. Unlike stable coronary artery disease (CAD) or chronic coronary syndrome (CCS), ACS patients have higher platelet reactivity and risk of recurrent ischemic events in the first year after PCI [5,6,7]. Recent SMART-DATE trial and meta-analysis had tried to shorten the duration of P2Y12 inhibitor in ACS patients, but they all failed [8,9]. The ischemic events were significantly increased once the duration of DAPT was shortened. Therefore, 12-months of DAPT is still strongly recommended for all ACS patients in the current guidelines if there is no specific contraindication for DAPT [10,11,12].

The P2Y12 receptor, a G-protein-coupled receptor (GPCR) coupled to the inhibitory G protein Gαi2, is a platelet ADP-receptor. The activation of platelet P2Y12 receptors by ADP leads to an inhibition of adenylyl cyclase and additional downstream events including the activation of phosphatidylinositol-3-kinase and the inhibition of Ras GTPase-activating protein 3 (RASA3) to promote GTPase Rap1b activity and integrin activation. These reactions eventually cause the amplification and stabilization of platelet aggregation [13]. By blocking P2Y12 receptors and ADP-induced platelet activation, P2Y12 inhibitors demonstrate potent antiplatelet effects. P2Y12 inhibitor monotherapy is a novel treatment strategy that shortens the duration of DAPT to 1–3 months and continues with a P2Y12 inhibitor instead of aspirin. The scientific rationale of this treatment strategy is to use a more potent antiplatelet agent to prevent recurrent ischemic events and avoid potential gastrointestinal side effects caused specifically by aspirin. This novel approach has been tested in several large randomized controlled studies, and nearly all of them had favorable outcomes [14,15,16,17]. Overall, P2Y12 inhibitor monotherapy could lower the risk of bleeding complications without increasing the ischemic events in the general population after PCI and stent implantation [18]. However, whether this novel approach could apply to ACS patients remains under debate,. In particular, the concern of a much higher risk of recurrent ischemic events in ACS patients than in stable CAD patients and the inconsistent antiplatelet effect of clopidogrel concern clinicians [19,20]. The purpose of this study is to evaluate the efficacy and safety of this novel approach among patients with ACS. We conducted a meta-analysis of randomized controlled trials that compared P2Y12 inhibitor monotherapy with 12-month DAPT in ACS patients who underwent PCI with stent implantation.

## 2. Results

### 2.1. Search Results and Characteristics of Included Trials

The results of the database search and study selection are shown in Figure 1. A total of 2289 records were identified from the databases and websites mentioned above. Of these, 74 full-text articles were reviewed, and 69 of them were excluded for not meeting the pre-specified inclusion criteria. In the end, five randomized controlled trials with a total of 21,034 patients were included in this meta-analysis. The main characteristics and outcomes of the included trials were summarized in Table 1. There were 10,556 patients who received standard 12-month DAPT, and 10,478 patients received P2Y12 inhibitor monotherapy after PCI and stent implantation. The ischemic and bleeding events of each trial are summarized in Table 2.

### 2.2. The Primary and Secondary Outcomes

The quantitative analysis of primary and secondary outcomes is shown in Figure 2. In ACS patients, P2Y12 inhibitor monotherapy did not increase the risk of MACCE as compared with standard 12-month DAPT (OR: 0.98, 95% CI: 0.86–1.13, *p* = 0.82, *I*^2^ = 41%, P_Heterogeneity_ = 0.15) (Figure 2A), but the risk of major or minor bleeding events was significantly lower in patients treated with P2Y12 inhibitor monotherapy (OR: 0.59, 95% CI: 0.46–0.75, *p* < 0.0001, *I*^2^ = 58%, P_Heterogeneity_ = 0.05) (Figure 2B). The quantitative analysis of NACE demonstrated a significantly favorable result for P2Y12 inhibitor monotherapy (OR: 0.82, 95% CI: 0.73–0.93, *p* = 0.002, *I*^2^ = 29%, P_Heterogeneity_ = 0.23) in patients with ACS after PCI (Figure 2C).

### 2.3. Subgroup Analysis of Different P2Y12 Inhibitors

The efficacy of P2Y12 inhibitor monotherapy was quite different among patients receiving ticagrelor from those receiving clopidogrel. Although statistically insignificant, patients receiving clopidogrel monotherapy were associated with a trend of higher risk of MACCE as compared with standard DAPT (OR: 1.50, 95% CI: 0.99–2.28, *p* = 0.06), whereas patients receiving ticagrelor monotherapy were associated with a favorable result (OR: 0.92, 95% CI: 0.78–1.09, *p* = 0.34) (Figure 3). Both clopidogrel monotherapy and ticagrelor monotherapy showed a similar reduction in the risk of bleeding events (OR: 0.46, 95% CI: 0.22–0.94; OR: 0.60, 95% CI: 0.44–0.83, respectively) (Figure 4). Overall, the NACE was significantly lower in ticagrelor monotherapy (OR: 0.79, 95% CI: 0.69–0.91, *p* = 0.001) as compared with standard DAPT. However, the NACE was no different between clopidogrel monotherapy and standard DAPT (OR: 1.14, 95% CI: 0.79–1.63, *p* = 0.49) (Figure 5). The SMART-CHOICE study was not included in the subgroup analysis because there was no available reported data on different P2Y12 inhibitors.

### 2.4. Extrapolatory Analysis of P2Y12 Inhibitor Monotherapy in Non-ACS Patients as Compared with ACS Patients

The primary efficacy and safety outcomes of P2Y12 inhibitor monotherapy in non-ACS versus non-ACS patients in these included clinical trials are summarized in Table 3. The quantitative analysis of the clinical outcomes in non-ACS patients is demonstrated in Figure 6. P2Y12 inhibitor monotherapy was associated with a favorable result of reducing major bleeding as compared with 12-month DAPT, but statistically insignificant (OR: 0.83, 95% CI: 0.66–1.05, *p* = 0.13). The 1-year rate of ischemic events was similar between P2Y12 inhibitor monotherapy and DAPT in both non-ACS (OR: 1.03, 95% CI: 0.85–1.25, *p* = 0.78).

### 2.5. Extraploartory Analysis of P2Y12 Inhibitor Monotherapy in STE-ACS Patients as Compared with NSTE-ACS Patients

The efficacy and safety outcomes of P2Y12 inhibitor monotherapy in STE-ACS versus NSTE-ACS patients in these included clinical trials were demonstrated in Figure 7 and Figure 8. The incidence of ischemic events was similar between P2Y12 inhibitor monotherapy and DAPT in both STE-ACS (OR: 1.13, 95% CI: 0.86–1.48, *p* = 0.38) and NSTE-ACS patients (OR: 0.89, 95% CI: 0.76–1.05, *p* = 0.17). P2Y12 inhibitor monotherapy significantly reduced bleeding events as compared with 12-month DAPT in both STE-ACS (OR: 0.68, 95% CI: 0.49–0.96, *p* = 0.03) and NSTE-ACS patients (OR: 0.60, 95% CI: 0.51–0.71, *p* < 0.0001). The SMART-CHOICE study was not included in the subgroup analysis because there was no available reported data on the different diagnoses of ACS.

### 2.6. Quality Assessment and Publication Bias

The overall risk of bias in selection, detection, and reporting bias was low. The detailed quality assessment and risk of bias assessment per study can be found in Appendix A. All studies in this meta-analysis were randomized controlled trials, but only TWILIGHT-ACS was double-blinded. There was no publication bias in all outcomes. The outcomes of included trials are distributed symmetrically in the funnel plot (Appendix A), and heterogeneity was low in all outcomes.

## 3. Materials and Methods

### 3.1. Data Sources and Searching

This meta-analysis was conducted following the guidelines of the Preferred Reporting Items for a Systematic review and Meta-analysis (PRISMA) and Cochrane Collaboration. Our protocol was registered on PROSPERO (international prospective register of systematic reviews) and is available online (www.crd.york.ac.uk/prospero, accessed on 29 March 2022, CRD42022312669). We searched PubMed, Embase, the Cochrane library database, ClinicalTrials.gov, and three other websites (www.tctmd.com, www.acc.org/cardiosourceplus, and www.escardio.org) from the earliest record to July 2022. The search terms used included: “P2Y12 inhibitor monotherapy”, “dual antiplatelet therapy”, “randomized trial”, “percutaneous coronary intervention”, “outcome”, and “acute coronary syndrome”. No language restriction was applied.

### 3.2. Study Selection

The inclusion criteria of the selected study were: (1) ACS patients who underwent PCI with stent implantation, (2) randomized controlled trial (or subgroup analysis of randomized controlled trial), (3) comparing P2Y12 inhibitor monotherapy to standard 12-month dual antiplatelet therapy, (4) follow up patients’ clinical outcomes for at least 12 months, and (5) the study reported the primary efficacy and safety outcomes of interest. The exclusion criteria were: (1) a non-randomized controlled trial, (2) studies not reporting the data of patients with ACS, and (3) ongoing studies or lack of clinical endpoints data, and studies not available in full text were also excluded.

All of the retrieved articles were screened by three reviewers to identify all of the potentially eligible studies.

### 3.3. Data Extraction and Clinical Outcomes

The baseline characteristics and outcome data of the included studies were independently extracted by multiple reviewers, and the discrepancy was resolved through negotiation. The primary efficacy outcome was major adverse cardiovascular and cerebrovascular events (MACCE), a composite of all-cause mortality, myocardial infarction, stent thrombosis, or stroke. The primary safety outcome was major or minor bleeding events. The secondary endpoint was net adverse clinical event (NACE), defined as a composite of major bleeding and adverse cardiac and cerebrovascular events. A separate subgroup analysis on the type of P2Y12 inhibitor was performed.

### 3.4. Assessment of Risk of Bias

The quality of each study is independently evaluated by the first and second authors (Wen-Han Feng and Yong-Chieh Chang) by using the Cochrane Collaboration tool. Discrepancies were solved by discussions with the corresponding author.

### 3.5. Data Synthesis and Analysis

All data were pooled to calculate the odds ratio (OR) and 95% confidence intervals by using a random-effects model. Between-trial heterogeneity was assessed by using an I^2^ test, and whether the value > 50% was regarded as considerable heterogeneity. Potential publication bias was examined by the visual inspection of funnel plots. Statistical significance is defined as *p*-value < 0.05. All analyses were performed using Review Manager (RevMan) software, version 5.4. (The Cochrane Collaboration, 2020.)

## 4. Discussion

The results from this meta-analysis of 21,034 patients from five randomized controlled trials demonstrated that P2Y12 inhibitor monotherapy could significantly lower the risk of bleeding complications without increasing the risk of ischemic events as compared with standard DAPT in ACS patients after PCI and stent implantation. The benefit of this novel approach was clearly demonstrated by the results of NACE. Moreover, this benefit was consistent in both NSTE-ACS and STE-ACS patients. These findings might challenge contemporary practice guideline recommendations of 12-month DAPT as the standard treatment for ACS patients after PCI.

In our exploratory analysis, the benefit of reducing bleeding events in P2Y12 inhibitor monotherapy was more pronounced among ACS patients as compared with non-ACS patients (relative risk reduction was 45% vs. 10%, respectively). This result may reflect the differences in underlying demographic and clinical characteristics between ACS patients and non-ACS patients. In the GLOBAL LEADERS study, non-ACS patients were older and had more co-morbidities (such as diabetes, peripheral vascular disease, chronic kidney disease, and prior MI) than ACS patients. Although ACS patients were younger and had fewer co-morbidities, they were still associated with a more significant reduction of bleeding events by the withdrawal of aspirin [25]. Similar findings were observed both in the TWILIGHT study and the STOPDAPT-2 study [22,26]. These observations also support the preferential benefit of applying P2Y12 inhibitor monotherapy to ACS patients other than non-ACS patients.

How to balance the risk of ischemic and bleeding events in treating ACS patients remains a crucial question. ACS patients have higher platelet reactivity and a higher risk of recurrent ischemic events than non-ACS patients [6]. Therefore, current guidelines recommend at least 12 months of DAPT in ACS patients, even without PCI and stent implantation [27,28]. Notably, the risk of bleeding events was also higher in ACS patients. A previous study had demonstrated that once the patients had post-PCI bleeding, the 2-year mortality risk was about eight times higher than for those without post-PCI bleeding [3].

Aspirin and P2Y12 inhibitors have different pharmacological pathways to inhibit the activation of platelets. Aspirin suppresses the generation of thromboxane A2 by acetylating cyclooxygenase-1. P2Y12 inhibitors block the P2Y12-dependent pathway by either directly blocking adenosine diphosphate (ADP)-induced signal transduction (ticagrelor) or blocking the binding of ADP to P2Y12 receptor (clopidogrel, prasugrel) [29]. DAPT was presumed to have additive inhibitory effects on platelet activation [30]. However, studies found that aspirin only had a slight additional inhibition of platelet aggregation when a P2Y12 inhibitor was used [31]. Ticagrelor monotherapy was demonstrated to have similar levels of inhibition for most platelet activation pathways as compared with DAPT (ticagrelor plus aspirin) in patients that underwent PCI [32]. Experimental studies also showed that the blockade of platelet P2Y12 receptor reduced the generation of thromboxane A2 induced by platelet agonists, and subsequently inhibited the effects of thromboxane A2-induced ADP release [33,34]. Moreover, the aspirin-induced gastrointestinal irritation/bleeding and the potential aspirin-resistance phenotype are the other two concerns. The literature review revealed the prevalence of aspirin resistance to be approximately 20–30% in patients with cardiovascular disease [35,36]. These patients are at a greater risk of clinical adverse cardiovascular events and mortality than those sensitive to aspirin treatment [36].

Different from previous published meta-analyses and reviews on P2Y12 inhibitor monotherapy after PCI [37,38,39,40,41], our meta-analysis for the first time included the latest published STOPDAPT-2 ACS trial and characterized the significant difference between ticagrelor versus clopidogrel monotherapy in ACS patients. This distinction is important given the growing application of P2Y12 inhibitor monotherapy in patients after PCI and stent implantation. There are three oral P2Y12 inhibitors currently available in clinical practice, and they carry very different pharmacological characteristics. Clopidogrel and prasugrel are prodrugs that require the metabolism to cause them to enter their active form and to exert their antiplatelet effects. In contrast, ticagrelor is a direct-acting drug with no effect on P2Y12 genetic polymorphism. Clopidogrel had a relatively slow onset and modest antiplatelet effect as compared with the other two P2Y12 inhibitors [42]. More importantly, the response to clopidogrel is variable, and a substantial portion of patients may have a poor response or even resistance to this drug [19,20]. Current guidelines recommend that ticagrelor and prasugrel are the preferred P2Y12 inhibitor in DAPT for ACS patients by the clinical trial results from PLATO and TRITON-TIMI 38, respectively, unless they are unavailable or cannot be tolerated [43,44]. This recommendation seems to be the same in P2Y12 inhibitor monotherapy. In a STOPDAPT-2 ACS study, clopidogrel monotherapy after 1–2 months of DAPT failed to reach the noninferiority of 12-month DAPT for a composite of cardiovascular and bleeding events (HR: 1.14, 95% CI, 0.80–1.62, *p* = 0.06 for noninferiority) [24]. Although the major bleeding events were reduced, the incidence of cardiovascular events significantly increased. It is worthy of note that the incidence of myocardial infarction was higher in the clopidogrel monotherapy group than in the 12-month DAPT group (1.59% vs. 0.85%, HR: 1.91, 95% CI: 1.06–3.44), and most of these were spontaneous myocardial infarctions (1.5% vs. 0.8%, HR: 2.03, 95% CI: 1.09–3.78). Furthermore, in the subgroup analysis of STOPDAPT-2 ACS, the incidence of major secondary cardiovascular endpoint (a composite of cardiovascular death, MI, definite stent thrombosis, and stroke) was significantly higher in STEMI patients treated with clopidogrel monotherapy as compared with DAPT (2.84% vs. 1.61%, HR: 1.80, 95% CI: 1.01–3.19, *p* = 0.04). Based on the above findings, significant attention should be paid to clopidogrel monotherapy in ACS patients, especially in the STEMI patients.

Prasugrel monotherapy was tested in several clinical studies [45,46,47], but none of them involved a randomized controlled trial. Therefore, they were excluded from our meta-analysis. The largest study of prasugrel monotherapy was the PENDULUM mono and registry study [47]. It was a prospective, observational cohort study. The results showed that prasugrel monotherapy could reduce major or minor bleeding (OR: 0.68, 95% CI: 0.47–0.98, *p* = 0.039) without increasing ischemic events (OR: 0.85, 95% CI: 0.61–1.19, *p* = 0.348) as compared with DAPT. Prasugrel monotherapy seems to be a potential alternative treatment strategy in patients with high-bleeding risk, but further study is required to prove its efficacy and safety.

The TICO study was the very first randomized study to show that P2Y12 inhibitor monotherapy with ticagrelor could have better outcomes as compared to standard DAPT in ACS patients, even including patients with STEMI. However, the case number was relatively small. Our meta-analysis provided a greater amount of data to evaluate the efficacy and safety of ticagrelor monotherapy in ACS patients, and the findings were very consistent. Our recent real-world observational study also supported the findings of the present meta-analysis by demonstrating ticagrelor monotherapy to be associated with a substantially lower cardiovascular risk as compared with clopidogrel monotherapy in ACS patients that underwent PCI [48]. Therefore, ticagrelor is the preferred choice of the P2Y12 inhibitor when applying P2Y12 inhibitor monotherapy in ACS patients after PCI.

There are several limitations in our study. First, most patients enrolled in these trials were implanted with newer-generation DES. It is unclear whether our findings could apply to first-generation DES or bare-metal stents. Second, baseline characteristics and the indications for PCI were not identical in these included trials. Of note, the TWILIGHT study only enrolled patients who were able to tolerate three months of DAPT without having a major adverse clinical event. Those who had major bleeding or recurrent ischemic events within 90 days after PCI were excluded. Third, only one randomized trial involved the analysis of clopidogrel monotherapy since there was no available data from the SMART-CHOICE study. Fourth, prasugrel monotherapy was not analyzed in our study. This is because prasugrel was used in only 4% of the enrolled patients in the SMART-CHOICE study, and the other four clinical trials were not using prasugrel in P2Y12 inhibitor monotherapy. Fifth, some of our included trials were not global studies. The TICO study and SMART-CHOICE were conducted exclusively in South Korea, whereas the STOPDAPT-2 ACS study was only conducted in Japan. Caution is needed in extrapolating these results outside of East Asian patients. Racial differences are important issues in antiplatelet therapy. Platelet aggregation is indeed a process that may depend upon race [49]. East Asians have a higher frequency of the CYP2C19 loss-of-function alleles, and tend to have a lower incidence of ischemic outcomes and a higher incidence of bleeding outcomes compared to Caucasians. Black individuals have a higher prevalence of CV risk factors, and higher thrombogenic, proinflammatory, and dysfunctional endothelial profiles than Caucasians [50]. Future studies are needed to explore the efficacy and safety of P2Y12 inhibitor monotherapy in ACS in different races.

## 5. Conclusions

Based on the results of our study, P2Y12 inhibitor monotherapy could significantly decrease bleeding events without increasing the risk of stent thrombosis or myocardial infarction in ACS patients. However, the type of P2Y12 inhibitor did matter. Compared with the standard DAPT, P2Y12 inhibitor monotherapy by ticagrelor, but not clopidogrel, carries a significantly lower NACE. We conclude that P2Y12 inhibitor monotherapy with ticagrelor is a favorable choice for ACS patients after PCI with stent implantation.

## Figures and Tables

**Figure 1 pharmaceuticals-16-00232-f001:**
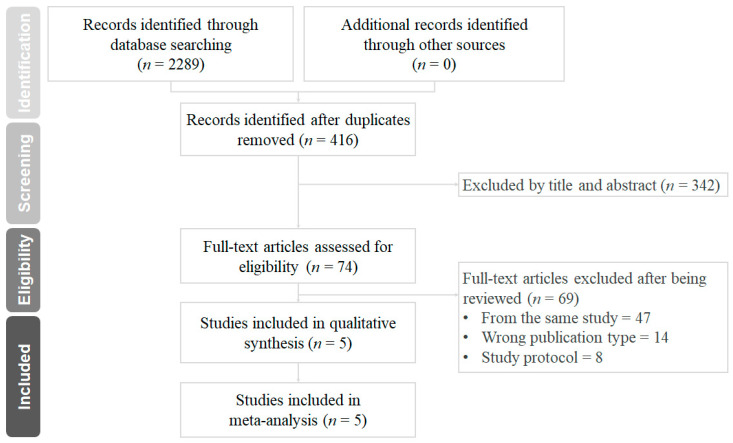
Preferred reporting items for systematic reviews and meta-analyses (PRISMA) diagram for the searching and identification of included studies.

**Figure 2 pharmaceuticals-16-00232-f002:**
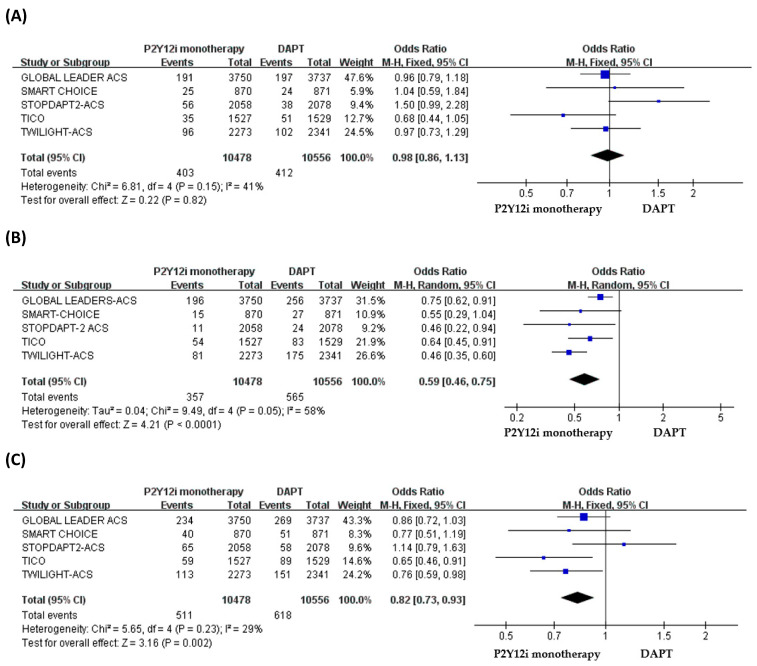
The efficacy and safety of P2Y12 inhibitor monotherapy in patients with ACS after PCI as compared with 12-month DAPT. (**A**) MACCE, (**B**) bleeding events, (**C**) NACE.

**Figure 3 pharmaceuticals-16-00232-f003:**
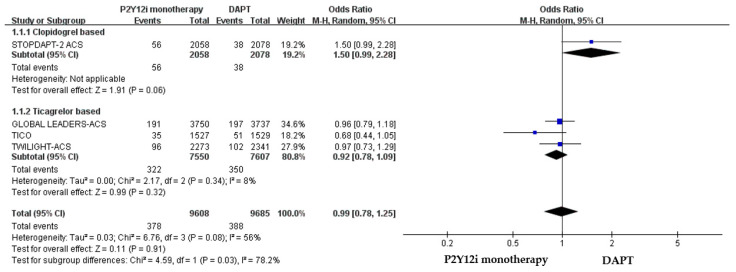
The efficacy outcomes (MACCE) of different P2Y12 inhibitor monotherapy as compared with 12-month DAPT in patients with ACS after PCI.

**Figure 4 pharmaceuticals-16-00232-f004:**
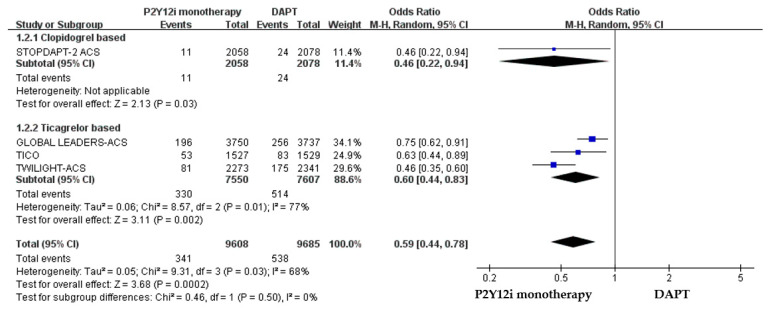
The safety outcomes (major or minor bleeding events) of different P2Y12 inhibitor monotherapy as compared with 12-month DAPT in patients with ACS after PCI.

**Figure 5 pharmaceuticals-16-00232-f005:**
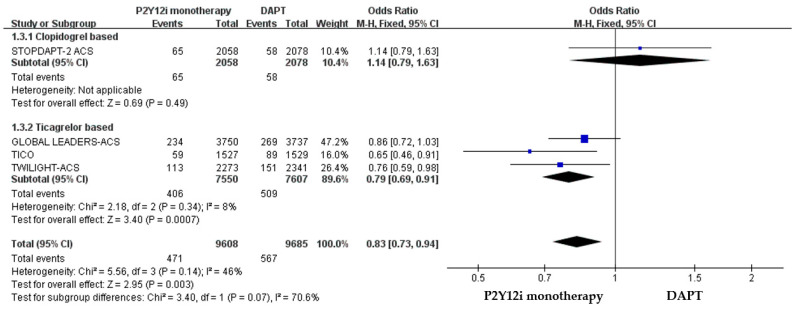
The NACE of different P2Y12 inhibitor monotherapy as compared with 12-month DAPT in patients with ACS after PCI.

**Figure 6 pharmaceuticals-16-00232-f006:**
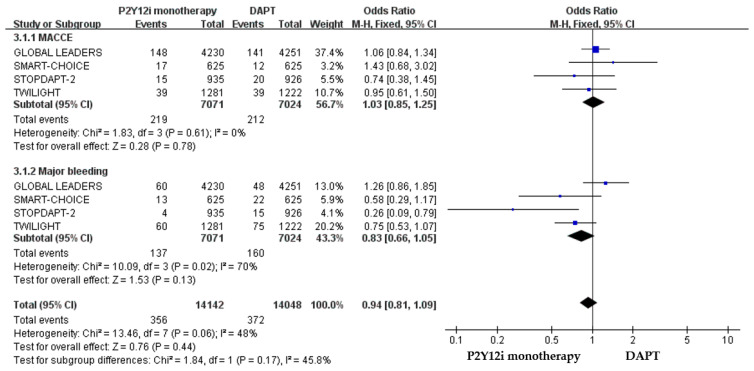
The efficacy and safety of P2Y12 inhibitor monotherapy in patients without ACS after PCI as compared with 12-month DAPT.

**Figure 7 pharmaceuticals-16-00232-f007:**
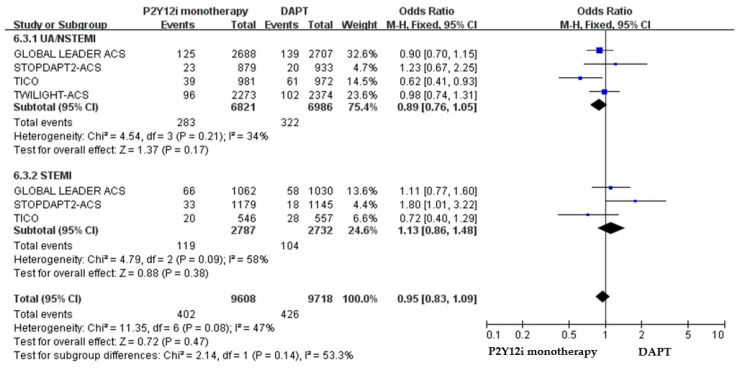
The efficacy outcomes (MACCE) of P2Y12 inhibitor monotherapy in STE-ACS patients and NSTE-ACS after PCI as compared with 12-month DAPT.

**Figure 8 pharmaceuticals-16-00232-f008:**
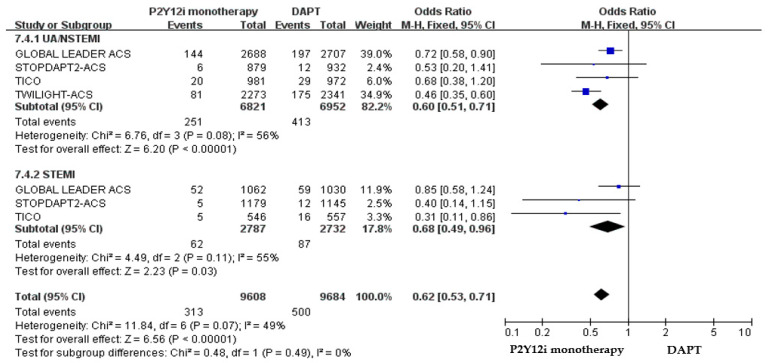
The safety outcomes (major or minor bleeding events) of P2Y12 inhibitor monotherapy as compared with 12-month DAPT in patients with STE-ACS and NSTE-ACS patients after PCI.

**Table 1 pharmaceuticals-16-00232-t001:** Clinical characteristics and outcomes of included randomized trials.

Clinical Trials	Global LEADERS ACS	Global LEADERS ACS	SMART- CHOICE	SMART- CHOICE	TWILIGHT ACS	TWILIGHT ACS	TICO	TICO	STOPDAPT-2 ACS	STOPDAPT-2 ACS
Year	2018	2018	2019	2019	2019	2019	2020	2020	2022	2022
Arm	Tica mono	DAPT	P2Y12i mono	DAPT	Tica mono	DAPT	Tica mono	DAPT	Clop mono	DAPT
DAPT months	1	12	3	12	3	12	3	12	1	12
Patients number	3750	3737	870	871	2273	2341	1527	1529	2078	2091
Age (mean)	64.5	64.6	64.4	64.4	64.2	64.2	61	61	67.0	66.6
Male	2880 (76.8)	2883 (77.1)	629 (72.3)	N/A	1693 (74.5)	1760 (75.2)	1204 (78.8)	1224 (80.0)	1631 (79.3)	1649 (79.4)
Prior MI	685 (18.3)	695 (18.6)	34 (3.9)	N/A	578 (25.4)	589 (25.2)	64 (4.2)	49 (3.2)	135 (6.6)	109 (5.3)
DM	809 (21.6)	795 (21.3)	318 (36.6)	N/A	810 (35.6)	804 (34.3)	418 (27.4)	417 (27.2)	608 (29.5)	621 (29.9)
STEMI	1062 (28.3)	1030 (27.6)	164 (18.9)	150 (17.2)	Excluded	Excluded	546 (35.7)	557 (36.4)	1179 (74.7)	1145 (72.8)
NSTEMI	1684 (44.9)	1689 (45.2)	239 (27.4)	230 (26.4)	2273 (100)	2341 (100)	539 (35.3)	488 (31.9)	399 (25.3)	427 (27.2)
Ischemic outcomes										
MACCE	191 (5.1)	197 (5.3)	25 (3.0)	24 (2.9)	96 (4.3)	102 (4.4)	35 (2.3)	51 (3.4)	56 (2.7)	38 (1.9)
All-cause death	59 (1.6)	75 (2.0)	12 (1.4)	N/A	22 (1.0)	34 (1.5)	16 (1.1)	23 (1.5)	28 (1.4)	19 (0.9)
MI	96 (2.6)	88 (2.4)	8 (0.9)	N/A	70 (3.1)	72 (3.1)	6 (0.4)	11 (0.7)	32 (1.6)	17 (0.9)
Stroke	28 (0.8)	26 (0.7)	6 (0.7)	N/A	11 (0.5)	6 (0.3)	8 (0.5)	11 (0.7)	15 (0.7)	11 (0.5)
Stent thrombosis	25 (0.7)	23 (0.6)	N/A	N/A	8 (0.4)	14 (0.6)	6 (0.4)	4 (0.3)	10 (0.5)	4 (0.2)
Bleeding outcomes										
Major or minor bleeding	196 (5.2)	256 (6.9)	15 (1.8)	27 (3.2)	81 (3.6)	175 (7.6)	53 (3.6)	83 (5.5)	11 (0.5)	24 (1.2)
Major bleeding	57 (1.5)	88 (2.4)	N/A	N/A	17 (0.8)	49 (2.1)	25 (1.7)	45 (3.0)	7 (0.3)	13 (0.6)
NACE	234 (6.2)	269 (7.2)	40 (4.6)	51 (5.9)	113 (5.0)	151 (6.5)	59 (3.9)	89 (5.9)	65 (3.2)	58 (2.8)

Values are N (%) unless otherwise indicated. DAPT: dual antiplatelet therapy, DM: diabetes mellitus, MACCE: major adverse cardiovascular and cerebrovascular events, MI: myocardial infarction, NACE: net adverse clinical events, N/A: not applicable, P2Y12i: P2Y12 inhibitor, STEMI: ST-elevation myocardial infarction, NSTEMI: non ST-elevation myocardial infarction.

**Table 2 pharmaceuticals-16-00232-t002:** The efficacy and safety outcomes of P2Y12 inhibitor monotherapy and DAPT in ACS patients of included trials.

	P2Y12i Monotherapy	DAPT	OR (95% CI)
GLOBAL LEADERS-ACS [21] (*n* = 7487)			
MACCE	191 (5.1)	197 (5.3)	0.96 (0.77–1.18)
Major or minor bleeding	196 (5.2)	256 (6.9)	0.84 (0.71–1.00)
NACE	234 (6.2)	269 (7.2)	0.87 (0.76–1.01)
SMART-CHOICE [15] (*n* = 1741)			
MACCE	25 (3.0)	24 (2.9)	1.06 (0.61–1.85)
Major or minor bleeding	15 (1.8)	27 (3.2)	0.56 (0.30–1.05)
NACE	40 (4.6)	51 (5.9)	0.77 (0.51–1.19)
TWILIGHT-ACS [22] (*n* = 4614)			
MACCE	96 (4.3)	102 (4.4)	0.97 (0.74–1.28)
Major or minor bleeding	81 (3.6)	175 (7.6)	0.47 (0.36–0.61)
NACE	113 (5.0)	151 (6.5)	0.76 (0.59–0.98)
TICO [23] (*n* = 3056)			
MACCE	35 (2.3)	51 (3.4)	0.69 (0.45–1.06)
Major or minor Bleeding	53 (3.6)	83 (5.5)	0.64 (0.45–0.90)
NACE	59 (3.9)	89 (5.9)	0.66 (0.48–0.92)
STOPDAPT-2 ACS [24] (*n* = 4136)			
MACCE	56 (2.7)	38 (1.9)	1.50 (0.99–2.26)
Major or minor Bleeding	11 (0.5)	24 (1.2)	0.46 (0.23–0.94)
NACE	65 (3.2)	58 (2.8)	1.14 (0.80–1.62)

Values are N (%) unless otherwise indicated. DAPT: dual antiplatelet therapy, MACCE: major adverse cardiac and cerebrovascular event, NACE: net adverse clinical event, OR: odds ratio, P2Y12i: P2Y12 inhibitor.

**Table 3 pharmaceuticals-16-00232-t003:** The efficacy and safety outcomes of P2Y12 inhibitor monotherapy in ACS versus non-ACS patients of included trials.

	ACS			Non-ACS		
	P2Y12i Monotherapy	DAPT	OR (95% CI)	P2Y12i Monotherapy	DAPT	OR (95% CI)
GLOBAL LEADERS (*n)*	3750	3737		4230	4251	
MACCE	191 (5.1)	197 (5.3)	0.96 (0.77–1.18)	148 (3.5)	141 (3.3)	1.06 (0.84–1.34)
Major bleeding	57 (1.5)	88 (2.3)	0.64 (0.46–0.90)	60 (1.4)	48 (1.1)	1.26 (0.86–1.85)
SMART-CHOICE (*n)*	870	871		625	625	
MACCE	25 (3.0)	24 (2.9)	1.06 (0.61–1.85)	17 (2.8)	12 (2.0)	1.43 (0.68–3.00)
Major or minor bleeding	15 (1.8)	27 (3.2)	0.56 (0.30–1.05)	13 (2.2)	22 (3.6)	0.59 (0.30–1.18)
TWILIGHT (*n)*	2273	2341		1281	1222	
MACCE	96 (4.3)	102 (4.4)	0.97 (0.74–1.28)	39 (3.1)	39 (3.2)	0.96 (0.61–1.49)
Major or minor bleeding	81 (3.6)	175 (7.6)	0.47 (0.36–0.61)	60 (4.8)	75 (6.2)	0.76 (0.54–1.06)
STOPDAPT-2 (*n)*	2058	2078		935	926	
MACCE	56 (2.7)	38 (1.9)	1.50 (0.99–2.26)	15 (1.6)	20 (2.2)	0.74 (0.38–1.45)
Major or minor Bleeding	11 (0.5)	24 (1.2)	0.46 (0.23–0.94)	4 (0.4)	15 (1.6)	0.26 (0.09–0.79)

Values are N (%) unless otherwise indicated. DAPT: dual antiplatelet therapy, MACCE: major adverse cardiac and cerebrovascular event, NACE: net adverse clinical event, OR: odds ratio, P2Y12i: P2Y12 inhibitor.

## Data Availability

The original contributions presented in this study are included in the article/Appendix A, further inquiries can be directed to the corresponding author.

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
