# Peer review of "P2Y12 Inhibitor Monotherapy versus Conventional Dual Antiplatelet Therapy in Patients with Acute Coronary Syndrome after Percutaneous Coronary Intervention: A Meta-Analysis"

_pharmaceuticals, 2023, doi:10.3390/ph16020232_

Round 1

Reviewer 1 Report

Feng and collaborators  performed a meta-analysis of data obtained from 21000 patients affected by different types  of coronary syndromes having   used P2Y2 inhibitors versus DAPT as anti-platelets aggregation therapy.

They conclude that monotherapy with  P2Y2 inhibitors significantly decrease the risk of bleeding while saving the benefit on major adverse cardio vascular events. They however raise caution as to  the  specific P2Y12 inhibitor molecule to be used in such monotherapy.

The criteria of the authors to select the clinical trials used in their analysis is interesting. However as many meta-analysis  about the use of P2Y12 inhibitors have been reported including very recent ones ( Heart J Cardiovasc Pharmacother. 2022 Dec 23), it would be more helpful and would make the current review more original and up-to-date to broaden the number of trials to be included.  For example patient with coronary bypass have not been included in any of the trials used.

It could be very helpful to include more trials to increase the number of women as well as to look at the effect of genetics ( patients from differents races). Platelet aggregation is indeed a process that may depend upon sex, race… (see for example Arteriosclerosis, Thrombosis, and Vascular Biology. 2014;34:2644–2650

The discussion is quite verbose and mostly re-describes the results. It could be shortened and more focused.

Author Response

Reviewer 1:

Feng and collaborators performed a meta-analysis of data obtained from 21000 patients affected by different types of coronary syndromes having used P2Y12 inhibitors versus DAPT as anti-platelets aggregation therapy. They conclude that monotherapy with P2Y12 inhibitors significantly decreases the risk of bleeding while saving the benefit on major adverse cardiovascular events. They however raise caution as to the specific P2Y12 inhibitor molecule to be used in such monotherapy.

1. The criteria of the authors to select the clinical trials used in their analysis is interesting. However, as many meta-analysis about the use of P2Y12 inhibitors have been reported including very recent ones (Heart J Cardiovasc Pharmacother. 2022 Dec 23), it would be more helpful and would make the current review more original and up-to-date to broaden the number of trials to be included.  For example patient with coronary bypass have not been included in any of the trials used.

Response:

  1. Many thanks for the reviewer’s suggestion. Our study keeps focusing on ACS patients after PCI. The antithrombotic therapy for patients after coronary artery bypass surgery is quite interesting and could be our next research objective.
  2. We appreciate you providing this latest published meta-analysis of P2Y12 inhibitor monotherapy for complex PCI (Eur Heart J Cardiovasc Pharmacother. 2022 Dec 23;pvac071. PMID: 36564015) to enrich our knowledge. It has been added to our reference in the DISCUSSION section. We also found that their study objectives were different from ours. They focused on patients after complex PCI, and we analyzed patients with ACS. Also, we emphasized the difference in P2Y12 monotherapy between ticagrelor versus clopidogrel.

2. It could be very helpful to include more trials to increase the number of women as well as to look at the effect of genetics (patients from different races). Platelet aggregation is indeed a process that may depend upon sex, race… (see for example Arteriosclerosis, Thrombosis, and Vascular Biology. 2014;34:2644–2650)

Response:

  1. Thanks for the reviewer’s kind advice. We totally agree with the reviewer’s comment that racial disparities and gender differences are both important issues in antiplatelet therapy. The influence of different races and gender on the efficacy and safety of P2Y12 inhibitor monotherapy needs further study. We added some discussion on these issues in our DISCUSSION section to indicate this point as a study limitation.
  2. Thank you for providing an informative research article (ATVB 2014; 34:2644-2650. PMID: 25278289) to enrich our knowledge. It was added to our reference in the DISCUSSION section.
  3. We analyzed the efficacy and safety of P2Y12 inhibitor monotherapy in male vs female patients with ACS after PCI. The results are shown in the figures below (please see the attached file), for our reviewer’s quick reference. P2Y12 inhibitor monotherapy offered comparable efficacy and safety results as compared with standard DAPT in both male and female gender.

3. The discussion is quite verbose and mostly re-describes the results. It could be shortened and more focused.

Response:

  1. Thanks for the reviewer’s suggestion. We polished our DISCUSSION section and made it more reader-friendly.

Reviewer 2 Report

This interesting article is devoted to the evaluation of monotherapy with P2Y12 inhibitors in comparison with conventional dual anti-platelet therapy in patients with acute coronary syndrome after percutaneous coronary intervention. The authors conducted a meta-analysis of clinical trials including a large number of patients. The article is of clinical interest.

Comment:

1.      The abstract and section 3.3 state "clopidogrel monotherapy was associated with a trend of higher risk of MACCE than standard DAPT (OR:1.50, 95% CI:0.99-2.28, p=0.06)." The data obtained by the authors indicate that there is no statistical significance of the differences. It is recommended to pay attention to the statistical significance of the differences and make appropriate changes in the text.

2. It would be useful to add information on future research perspectives, given the limitations of the current study.

3. Literature references in the text are after a period in a sentence (e.g., line 58 "...decades.[1,2]". This needs to be corrected.

Author Response

Reviewer 2:

This interesting article is devoted to the evaluation of monotherapy with P2Y12 inhibitors in comparison with conventional dual anti-platelet therapy in patients with acute coronary syndrome after percutaneous coronary intervention. The authors conducted a meta-analysis of clinical trials including a large number of patients. The article is of clinical interest.

Comment:

1. The abstract and section 3.3 state "clopidogrel monotherapy was associated with a trend of higher risk of MACCE than standard DAPT (OR:1.50, 95% CI:0.99-2.28, p=0.06)." The data obtained by the authors indicate that there is no statistical significance of the differences. It is recommended to pay attention to the statistical significance of the differences and make appropriate changes in the text.

Response:

1. We appreciate the reviewer’s careful reminder and agree with the reviewer’s comment. We apologize for the imprecise words and descriptions in our manuscript. The abstract, section 3.3, and the conclusion were all revised.

2. It would be useful to add information on future research perspectives, given the limitations of the current study.

Response:

1. Thanks for the reviewer’s suggestion. We added some discussion on future research perspectives in our DISCUSSION section (page 13.)

3. Literature references in the text are after a period in a sentence (e.g., line 58 "...decades.[1,2]". This needs to be corrected.

Response:

1. We thank you for the kind reminder. We have carefully corrected it and rechecked all the formats of citations.

Round 2

Reviewer 1 Report

The authors did not expand the cohort of patients to be included in the pharmacological trial. This is unfoturnate.  Indeed the P2Y12 inhibitor class of pharmacological agents is   the focus of many surveys. This suggests that   they are very used drugs and thus require many investigations as to the clinical situations  for which they can be efficient. 

Another review has been published (Timing, Selection, Modulation, and Duration of P2Y12 Inhibitors for Patients With Acute Coronary Syndromes Undergoing PCI, J Am Coll Cardiol Intv, 16 (1) 1–18) and shoul be at least quoted 

Author Response

The authors did not expand the cohort of patients to be included in the pharmacological trial. This is unfortunate. Indeed the P2Y12 inhibitor class of pharmacological agents is the focus of many surveys. This suggests that they are very used drugs and thus require many investigations as to the clinical situations for which they can be efficient. 

Another review has been published (Timing, Selection, Modulation, and Duration of P2Y12 Inhibitors for Patients With Acute Coronary Syndromes Undergoing PCI, J Am Coll Cardiol Intv. 2023 Jan, 16 (1) 1–18) and should be at least quoted 

Response:

  1. Thank you for providing the latest review article (Am Coll Cardiol Intv. 2023 Jan, 16 (1) 1–18) to enrich our knowledge. It was added to our reference in the DISCUSSION section.

Reviewer 2 Report

The authors made changes to the article that improved its quality.

Round 3

Reviewer 1 Report

the referre regrets that the authors did not increase the cohorts of patients to be included in the pharmacological trial to broaden the spectrum of patients to be potentially  treated by the purinergic receptor antagonist